# Rapid Exploration for Open-World Navigation with Latent Goal Models

**Dhruv Shah**[1], **Benjamin Eysenbach**[2], **Nicholas Rhinehart**[1], **Sergey Levine**[1]
[1]UC Berkeley    [2]Carnegie Mellon University

[sites.google.com/view/recon-robot](sites.google.com/view/recon-robot)

**Abstract:** We describe a robotic learning system for autonomous exploration and navigation in diverse, open-world environments. At the core of our method is a learned latent variable model of distances and actions, along with a non-parametric topological memory of images. We use an information bottleneck to regularize the learned policy, giving us (i) a compact visual representation of goals, (ii) improved generalization capabilities, and (iii) a mechanism for sampling feasible goals for exploration. Trained on a large offline dataset of prior experience, the model acquires a representation of visual goals that is robust to task-irrelevant distractors. We demonstrate our method on a mobile ground robot in open-world exploration scenarios. Given an image of a goal that is up to 80 meters away, our method leverages its representation to explore and discover the goal in under 20 minutes, even amidst previously-unseen obstacles and weather conditions.

## 1 Introduction

Robustness is a key challenge in learning to navigate diverse, real-world environments. A robotic learning system must be robust to the difference between an offline training dataset and the real world (i.e., it must generalize), be robust to non-stationary changes in the real world (i.e., it must ignore visual distractors), and be equipped with mechanisms to actively explore to gather information about traversability. Different environments may exhibit similar physical structures, and these similarities can be used to accelerate exploration of *new* environments. Learning-based methods provide an appealing approach for learning a representation of this shared structure using prior experience.

In this work, we consider the problem of navigating to a user-specified goal in a previously unseen environment. The robot has access to a large and diverse dataset of experience from *other* environments, which it can use to learn general navigational affordances. Our approach to this problem uses an information bottleneck architecture to learn a compact representation of goals. Learned from prior data, this latent goal model encodes prior knowledge about perception, navigational affordances, and short-horizon control. We use a non-parametric memory to incorporate experience from the new environment. Combined, these components enable our system to learn to navigate to goals in a new environment after only a few minutes of exploration.

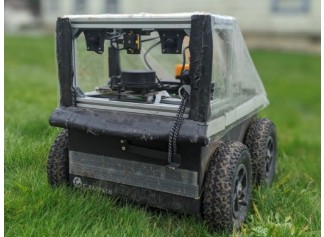

**Figure 1:** We demonstrate RECON on a Clearpath Jackal.

The primary contribution of this work is a method for exploring novel environments to discover user-specified goals. Our method operates directly on a stream of image observations, without relying on structured sensors or geometric maps. An important part of our method is a compressed representation of goal images that simultaneously affords robustness while providing a simple mechanism for exploration. Such a representation allows us, for example, to specify a goal image at one time of day, and then navigate to that same place at a different time of day: despite variation in appearance, the latent goal representations must be sufficiently close that the model can produce the correct actions. Robustness of this kind is critical in real-world settings, where the appearance of landmarks can change significantly with times of day and seasons of the year.

5th Conference on Robot Learning (CoRL 2021), London, UK.

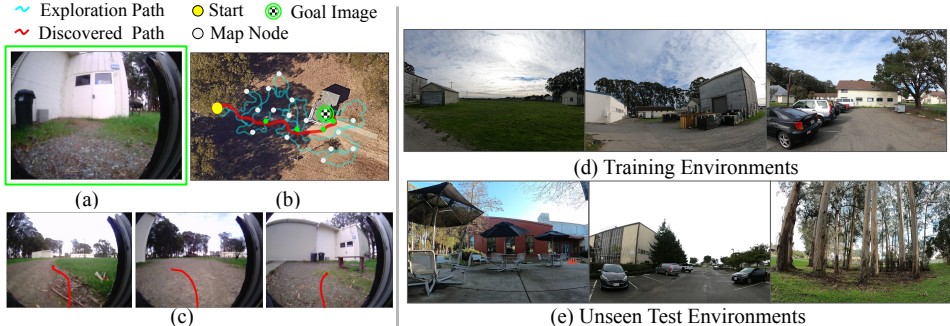

**Figure 2: System overview:** Given a goal image *(a)*, RECON explores the environment *(b)* by sampling prospective *latent* goals and constructing a topological map of images (white dots), operating only on visual observations. After finding the goal *(c)*, RECON can reuse the map to reach arbitrary goals in the environment (red path in *(b)*). RECON uses data collected from diverse training environments *(d)* to learn navigational priors that enable it to quickly explore and learn to reach visual goals a variety of unseen environments *(e)*.

We demonstrate our method, **R**apid **E**xploration **C**ontrollers for **O**utcome-driven **N**avigation (RE-CON), on a mobile ground robot (Fig. 1) and evaluate against 6 competitive baselines spanning over 100 hours of real-world experiments in 8 distinct open-world environments (Fig. 2). Our method can discover user-specified goals up to 80m away after just 20 minutes of interaction in a new environment. We also demonstrate robustness in the presence of visual distractors and novel obstacles. We make this dataset publicly available as a source of real-world interaction data for future resesarch.

## 2   Related Work

Exploring a new environment is often framed as the problem of efficient mapping, posed in terms of information maximization to guide the robot to uncertain regions of the environment. Some prior exploration methods use local strategies for generating control actions for the robots [1–4], while others use use global strategies based on the frontier method [5–7]. However, building high-fidelity geometric maps can be hard without reliable depth information. Further, such maps do not encode semantic aspects of traversability, e.g., tall grass is traversable but a wire fence is not.

Inspired by prior work [8–12], we construct a topological map by learning a distance function and a low-level policy. We estimate distances via supervised regression and learn a local control policy via goal-conditioned behavior cloning [13, 14]. However, these prior methods do not describe how to learn to navigate in *new*, unseen environments. We equip RECON with an explicit mechanism for exploring new environments and transferring knowledge across environments.

Well-studied methods for exploration in reinforcement learning (RL) utilize a novelty-based bonus, computed from a predictive model [15–21], information gain [22, 23], or methods based on counts, densities, or distance from previously-visited states [24–26]. However, these methods learn to reason about the novelty of a state only after visiting it. Recent works [27, 28] improve upon this by predicting explorable areas for interesting parts of the environment to accelerate visual exploration. While these methods can yield state-of-the-art results in simulated domains [29, 30], they come at the cost of high sample complexity (over 1M samples) and are infeasible to train in open-world environments without a simulated counterpart. Instead, our method enables the robot to explore an environment from scratch in just 20 minutes, using prior experience from other environments.

The problem of reusing experience across tasks is studied in the context of meta-learning [31–33] and transfer learning [34–38]. Our method uses an information bottleneck [39], which serves a dual purpose: first, it provides a representation that can aid the generalization capabilities of RL algorithms [40, 41], and second, it serves as a measure of task-relevant uncertainty [42], allowing us to incorporate prior information for proposing goals that are functionally-relevant for learning control policies in the new environment.

The problem of learning goal-directed behavior has been studied extensively using RL [43–46] and imitation learning (IL) [13, 14, 47–50]. Our method builds upon prior goal-conditioned IL methods to solve a slightly different problem: how to reach goals in a *new* environment. Once placed in a new environment, our method explores by carefully choosing which goals to visit, inspired by prior

work [51–55]. Unlike these prior methods, however, our method makes use of previous experience in *different* environments to accelerate learning in the current environment.

## 3   Problem Statement and System Overview

We consider the problem of goal-directed exploration for visual navigation in novel environments: a robot is tasked with navigating to a goal location $G$, given an image observation $o_g$ taken at $G$. Broadly, this consists of three separate stages: (1) learning from offline data, (2) building a map in a new environment, and (3) navigating to goals in the new environment. We model the task of navigation as a Markov decision process with time-indexed states $s_t \in \mathcal{S}$ and actions $a_t \in \mathcal{A}$. We *do not assume* the robot has access to spatial localization or a map of the environment, or access to the system dynamics. We use videos of robot trajectories in a variety of environments to learn general navigational skills and build a compressed representation of the perceptual inputs, which can be used to guide the exploration of novel environments. We make no assumption on the nature of the trajectories: they may be obtained by human teleoperation, self-exploration, or as a result of a preset policy. These trajectories need not exhibit intelligent behavior. Since the robot only observes the world from a single on-board camera and does not run any state estimation, our system operates in a partially observed setting. Our system commands continuous linear and angular velocities.

### 3.1   Mobile Robot Platform

We implement RECON on a Clearpath Jackal UGV platform (see Fig. 1). The default sensor suite consists of a 6-DoF IMU, a GPS unit for approximate global position estimates, and wheel encoders to estimate local odometry. In addition, we added a forward-facing 170° field-of-view RGB camera and an RPLIDAR 2D laser scanner. Inside the Jackal is an NVIDIA Jetson TX2 computer. The GPS and laser scanner can become unreliable in some environments [56], so we use them solely as safety controllers during data collection. Our method operates only using images taken from the onboard RGB camera, without other sensors or ground-truth localization.

### 3.2   Self-Supervised Data Collection & Labeling

Our aim is to leverage data collected in a wide range of different environments to enable the robot to discover and learn to navigate to novel goals in novel environments. We curate a dataset of self-supervised trajectories collected by a time-correlated random walk in diverse real-world environments (see Fig. 2 (d,e)). This data was collected over a span of 18 months and exhibits significant variation in appearance due to seasonal and lighting changes. We make this dataset publicly available[1] and provide further details in Appendix A.

## 4   RECON : A Method for Goal-Directed Exploration

Our objective is to design a robotic system that uses visual observations to efficiently discover and reliably reach a target image in a previously unseen environment. RECON consists of two components that enable it to explore new environments. The first is an uncertainty-aware, context-conditioned representation of goals that can quickly adapt to novel scenes. The second component is a topological map, where nodes represent egocentric observations and edges are the predicted distance between them, constructed incrementally from frontier-based exploration, maintaining a compact memory of the target environment.

### 4.1   Learning to Represent Goals

Our method learns a compact representation of goal images that is robust to task-irrelevant factors of variation. We learn this representation using a variant of the information bottleneck architecture [42, 57]. We use a context-conditioned representation of goals to learn a control policy in the target environment (Fig. 3 describes the graphical model). Letting $I(\cdot; \cdot)$ denote mutual information, the objective in Eq. 1 encourages the model to compress the incoming goal image $o_g$ into a repre-

---

[1]Available for download at `sites.google.com/view/recon-robot/dataset`.

sentation $z_t^g$ conditioned on the current observation $o_t$ that is predictive of the best action $a_t^g$ and the temporal distance $d_t^g$ to the goal (upper-case denotes random variables):

$$I\big((A_t^g, D_t^g); Z_t^g \mid o_t\big) - \beta I(Z_t^g; O_g \mid o_t) \tag{1}$$

Following [42], we approximate the intractable objective in Eq. 1 with a variational posterior and decoder (an upper bound), resulting in the maximization objective:

$$L = \frac{1}{|\mathcal{D}|} \sum_{(o_t, o_g, a_t^g, d_t^g) \in \mathcal{D}} \mathbb{E}_{p_\phi(z_t^g \mid o_g, o_t)} \left[\log q_\theta\big(a_t^g, d_t^g \mid z_t^g, o_t\big)\right] - \beta \mathrm{KL}\left(p_\phi(\cdot \mid o_g, o_t) || r(\cdot)\right) \tag{2}$$

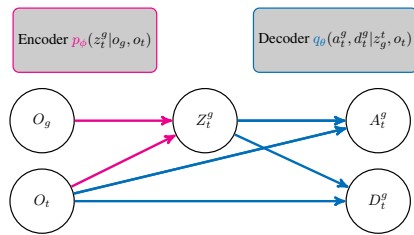

where we define the prior $r(z_t^g) \triangleq \mathcal{N}(0, I)$ and $\mathcal{D}$ is a dataset of trajectories characterized by $(o_t, o_g, a_t^g, d_t^g)$ quadruples. The first term measures the model's ability to predict actions and distances from the encoded representation, and the second term measures the model's compression of incoming goal images.

As the encoder $p_\phi$ and decoder $q_\theta$ are conditioned on $o_t$, the representation $z_t^g$ only encodes information about *relative* location of the goal from the context – this allows the model to represent *feasible* goals. If, instead, we had

**Figure 3:** Graphical model of actions and distances

a typical VAE (in which the input images are autoencoded), the samples from the prior over these representations would not necessarily represent goals that are reachable from the current state. This distinction is crucial when exploring *new* environments, where most states from the training environments are not valid goals.

## 4.2 Goal-Directed Exploration with Topological Memory

The second component of our system is a topological memory constructed incrementally as the robot explores a new environment. It provides an estimate of the exploration frontier as well as a map that the robot can use to later navigate to arbitrary goals. To build this memory, the robot uses the model from the previous section to propose *subgoals* for data collection. Note that this is done in the exploration phase and have a latent goal model pre-trained on the offline dataset. Given a subgoal, our algorithm (Alg. 1) proceeds by executing actions towards the subgoal for a fixed number of timesteps (Alg. 1 L12). The data collected during subgoal navigation expands the topological memory (Alg. 1 L14) and is used to fine-tune the model (Alg. 1 L15). Thus, the task of efficient exploration is reduced to the task of choosing subgoals.

---

**Algorithm 1** *RECON for Exploration:* RECON takes as input an encoder $p_\phi$, a decoder $q_\theta$, prior $r$, the current observation $o_t$ and goal observation $o_g$. $\delta_1, \delta_2, \epsilon, \beta \in \mathbb{R}_+$; $H, \gamma \in \mathbb{N}$ are hyperparameters.

---

1: **function** RECON $(q_\theta, p_\phi, r, o_t, o_g; \delta_1, \delta_2, \epsilon, \beta, \gamma, H)$
2:     $\mathcal{G} \leftarrow \emptyset, \mathcal{D} \leftarrow \emptyset$                                    ▷ *Initialize graph and data*
3:     **while** not reached goal $[\bar{d}_t^g < \delta_1]$ **do**                        ▷ *Continue while not at goal*
4:         $o_n \leftarrow \text{LeastExploredNeighbor}(\mathcal{G}, o_t; \delta_2)$
5:         $z_t^g \sim p_\phi(z \mid o_t, o_g)$                                        ▷ *Encode relative goal*
6:         **if** goal is feasible $[r(z_t^g) > \epsilon]$ **then**
7:             $z_t^w \leftarrow z_t^g$                                              ▷ *Will go to the goal*
8:         **else if** robot at frontier $[\bar{d}_t^n < \delta_1]$ **then**
9:             $z_t^w \sim r(z)$                                              ▷ *Will explore from frontier*
10:         **else**
11:             $z_t^w \sim p_\phi(z \mid o_t, o_n)$                                    ▷ *Will go to frontier*
12:         $\mathcal{D}_w, o_t \leftarrow \text{SubgoalNavigate}(z_t^w; H)$
13:         $\mathcal{D} \leftarrow \mathcal{D} \cup \mathcal{D}_w$
14:         $\text{ExpandGraph}(\mathcal{G}, o_t)$
15:         Step $L(\phi, \theta; \mathcal{D}, \beta)$ for $\gamma$ epochs                            ▷ Eq. 2
16:     **return** networks $p_\phi, q_\theta$ and graph $\mathcal{G}$

---

Subgoals are represented by latent variables in our model, which may either come from the posterior $p_\phi(z|o_t, o_g)$, or from the prior $r(z)$. Given a subgoal $z$ and observation $o_t$, the model decodes it into an action and distance pair $q(a_t^g, d_t^g|z, o_t)$; the action is used to control the robot towards the goal, and the distance is used to construct edges in the topological graph. The choice of intermediate subgoal to navigate toward at any step is based on the robot's estimate of the goal reachability and its proximity to the frontier. To determine the frontier of the graph, we track the number of times each node in the graph was selected as the navigation goal; nodes with low counts are considered to

**Algorithm 2** *RECON for Goal-Reaching:* After exploration, RECON uses the topological graph $\mathcal{G}$ to quickly navigate towards the goal $o_g$.

1: **procedure** GoalNavigate($\mathcal{G}, o_t, o_g; H$)
2:     $v_t \leftarrow$ AssociateToVertex($\mathcal{G}, o_t$)
3:     $v_g \leftarrow$ AssociateToVertex($\mathcal{G}, o_g$)
4:     $(v_t, \ldots, v_g) \leftarrow$ ShortestPath($\mathcal{G}, v_t, v_g$)
5:     **for** $v \in (v_t, \ldots, v_g)$ **do**
6:         $z \leftarrow p_\phi(z \mid o_t, o_g = v.\text{o})$
7:         $\mathcal{D}_w, o_t \leftarrow$ SubgoalNavigate($z; H$)

be on the frontier. In the following, we use $\bar{z}_t^g$ to denote the mean of the encoder $p_\phi(z \mid o_t, o_g)$, and $\bar{d}_t^g$ to denote the distance component of the mean of the decoder $q_\theta(a_t, d_t \mid \bar{z}_t^g, o_t)$ (i.e., the predicted number of time steps from $o_t$ to $\bar{z}_t^g$). The choice of subgoal at each step is made as follows:

**(i) Feasible Goal:** The robot believes it can reach the goal and adopts the representation of the goal image as the subgoal (Alg. 1 L7). The robot's confidence in reaching the goal is based on the probability of the current goal embedding $z_t^g$ under the prior $r(z)$. Large $r(z_t^g)$ implies the relationship between the observation $o_t$ and the goal $o_g$ is *in-distribution*, suggesting that the model's estimates of the distances is reliable – intuitively, this means that the model is confident about the distance to $o_g$ and can reach it.

**(ii) Explore at Frontier:** The robot is at the "least-explored node" (frontier) $o_n$ and explores by sampling a random conditional subgoal latent $z_t^w$ from the prior (Alg. 1 L9). The robot determines whether it is at the frontier based on the distance (estimated by querying the model) to its "least explored neighbor" $\bar{d}_t^n$ – the node in the graph within a distance threshold ($\delta_2$) of the current observation that has the lowest visitation count. If the distance to this node $\bar{d}_t^n$ is low (threshold $\delta_1$), then the robot is at the frontier.

**(iii) Go to Frontier:** The robot adopts its "least-explored neighbor" $o_n$ as a subgoal (Alg. 1 L11).

The SubgoalNavigate function rolls out the learned policy for a fixed time horizon $H$ to navigate to the desired subgoal latent $z_t^w$, by querying the decoder $q_\theta(a_t, d_t | z_t^w, o_\tau)$ with a fixed subgoal latent. The endpoint of such a rollout is used to update the visitation counts in the graph $\mathcal{G}$. At the end of each trajectory, the ExpandGraph subroutine is used to update the edge and node sets $\{\mathcal{E}, \mathcal{V}\}$ of the graph $\mathcal{G}$ to update the representation of the environment. We provide the pseudocode for these subroutines in Appendix B.1. We also share broader implementation details including choice of hyperparameters, model architectures and training details in Appendix B.2.

### 4.3 System Summary

RECON uses the latent goal model and topological graph to quickly explore new environments and discovers user-specified goals. Our complete system consists of three stages:

A) *Prior Experience:* The goal-conditioned distance and action model (Sec. 4.1) is trained using experience from previously visited environments. Supervision for training our model is obtained by using time steps as a proxy for distances and a relabeling scheme (Appendix A).

B) *Exploring a Novel Environment:* When placed in a new environment, RECON uses a combination of frontier-based exploration and latent goal-sampling with the learned model. The learned model is also fine-tuned to this environment. These steps are summarized in Alg. 1 and Sec. 4.2.

C) *Navigating an Explored Environment:* Given an explored environment (represented by a topological graph $\mathcal{G}$) and the model, RECON uses $\mathcal{G}$ to navigate to a goal image by planning a path of subgoals through the graph. This process is summarized in Alg. 2.

## 5 Experimental Evaluation

We designed our experiments to answer four questions:

| Method | Expl. Time (mm:ss) ↓ | Nav. Time (mm:ss) ↓ | SCT [58] ↑ |
|---|---|---|---|
| PPO + RND [18] | 21:18 | 00:47 | 0.22 |
| InfoBot [41] | 23:36 | 00:48 | 0.21 |
| Active Neural SLAM (ANS) [21] | 21:00 | 00:45 | 0.33 |
| ViNG [11] | 19:48 | 00:34 | 0.60 |
| Ours + Episodic Curiosity (ECR) [20] | 14:54 | 00:31 | 0.73 |
| **RECON (Ours)** | **09:54** | **00:26** | **0.92** |

**Table 1: Exploration and goal reaching performance:** Exploring 8 real-world environments, RE-CON reaches the goal 50% faster than the best baseline (ECR). ANS takes up to 2x longer to find the goal and NTS [27] fails to find the goal in every environment. On subsequent traversals, RECON navigates to the goal 20–85% faster than other baselines, and exhibits > 30% higher weighted success.

**Q1.** How does RECON compare to prior work for visual goal discovery in novel environments?

**Q2.** After exploration, can RECON leverage its experience to navigate to the goal efficiently?

**Q3.** What is the range of perturbations and non-stationary elements to which RECON is robust?

**Q4.** How important are the various components of RECON, such as sampling from an information bottleneck and non-parametric memory, to its performance?

## 5.1 Goal-Directed Exploration in Novel Environments

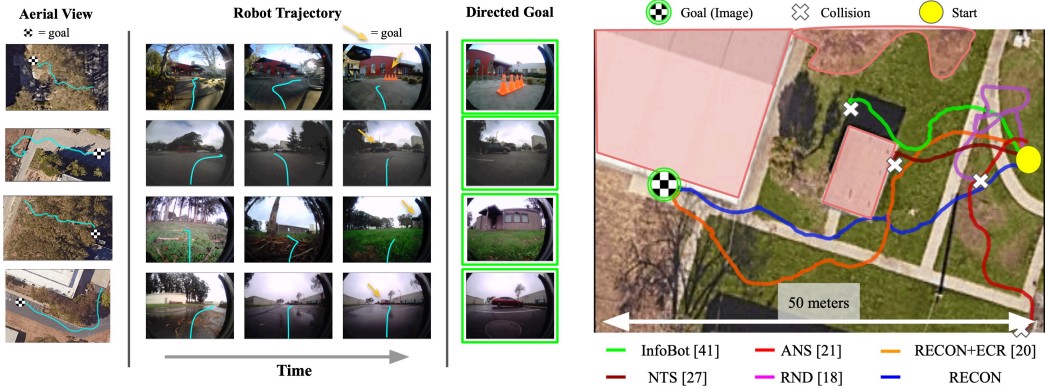

**Figure 4: Visualizing goal-reaching behavior of the system:** *(left)* Example trajectories to goals discovered by RECON in *previously unseen* environments. *(right)* Policies learned by the different methods in one such environment. Only RECON and ECR reach the goal successfully, and RECON takes the shorter route.

We perform our evaluation in a diverse variety of outdoor environments (examples in Fig. 2), including parking lots, suburban housing, sidewalks, and cafeterias. We train our self-supervised navigation model using an offline navigation dataset (Sec.3.2) collected in a distinct set of training environments, and evaluate our system's ability to discover user-specified goals in previously unseen environments. We compare RECON to five baselines, each trained on the same 20 hours of offline data as our method, and finetuned in the target environment with online interaction.

1. **PPO + RND:** Random Network Distillation (RND) is a widely used prediction bonus-based exploration strategy in RL [18], which we use with PPO [59, 60]. This comparison is representative of a frequently used approach for exploration in RL using a novelty-based bonus.

2. **InfoBot:** An offline variant of InfoBot [41], which uses goal-conditioned information bottleneck, analogous to our method, but does not use the non-parametric memory.

3. **Active Neural SLAM (ANS):** A popular indoor navigation approach based on metric spatial maps proposed for coverage-maximizing exploration [21]. We adapt it to the goal-directed task by using the distance function from RECON to detect when the goal is nearby.

4. **Visual Navigation with Goals (ViNG):** A method that uses random action sequences to explore and incrementally build a topological graph without reasoning about visitation counts [11].

5. **Episodic Curiosity (ECR):** A method that executes random action sequences at the frontier of a topological graph for exploration [20]. We implement this as an *ablation* of our method that samples random action sequences, rather than rollouts to sampled goals (Alg. 1 Line 7).

We evaluate the ability of RECON to discover visually-indicated goals in 8 *unseen* environments and navigate to them repeatedly. For each trial, we provide an RGB image of the desired target (one per environment) to the robot and report the time taken by each method to *(i)* discover the desired goal (**Q1**), and *(ii)* reliably navigate to the discovered goal a second time using prior exploration (**Q2**). Additionally, we quantify navigation performance using Success weighed by Completion Time (SCT), a success metric that takes into account the agent's dynamics [58]. We show quantitative results in Table 1, and visualize sample trajectories of RECON and the baselines in Fig. 4.

RECON outperforms all the baselines, discovering goals that are up to 80m away in under 20 minutes, including instances where no other baseline can reach the goal successfully. RECON+ECR and ViNG discover the goal in only the easier environments, and take up to 80% more time to discover the goal in those environments. RND, InfoBot and ANS are able to discover goals that are up to 25m away but fails to discover more distant goals, likely because using reinforcement learning for fine-tuning is data-inefficient. We exclude reporting metrics on NTS, which fails to successfully explore any environment, likely due to overfitting to the offline trajectories. Indeed, the simulation experiments reported in each of these online algorithms require upwards of 1M timesteps to adapt to new environments [21, 27, 41]. We attribute RECON's success to the context-conditioned sampling strategy (described in Sec. 4.1), which proposes goals that can accelerate the exploration of new environments.

We then study RECON's ability to quickly reach goals after initial discovery. Table 1 shows that RECON variants are able to quickly recall a feasible

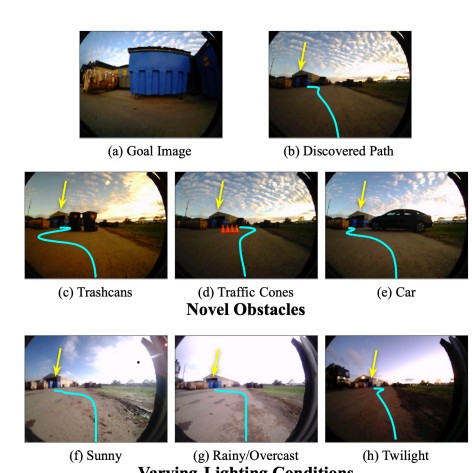

(a) Goal Image   (b) Discovered Path

(c) Trashcans  (d) Traffic Cones  (e) Car
**Novel Obstacles**

(f) Sunny  (g) Rainy/Overcast  (h) Twilight
**Varying Lighting Conditions**

**Figure 5: Exploring non-stationary environments:** The learned representation and topological graph is robust to visual distractors, enabling reliable navigation to the goal under novel obstacles *(c–e)* and appearance changes *(f–h)*.

path to the goal. These methods create a compact topological map from experience in the target environment, allowing them to quickly reach previously-seen states. The other baselines are unsuccessful at recalling previously seen goals for all but the simplest environments. Fig. 4 shows an aerial view of the paths recalled by various methods in one of the environments. Only the RECON variants are successfully able to navigate to the checkerboard goal; all other baselines result in collisions in the environment. Further, RECON discovers a shorter path to the goal and takes 30% less time to navigate to it than ECR ablation.

## 5.2 Exploring Non-Stationary Environments

Outdoor environments exhibit non-stationarity due to dynamic obstacles, such as automobiles and people, as well as changes in appearance due to seasons and time of day. Successful exploration and navigation in such environments requires learning a representation that is invariant to such distractors. This capability is of central interest when using a non-parametric memory: for the topological map to remain valid when such distractors are presented, we must ensure the invariance of the learned representation to such factors (**Q3**).

To test the robustness of RECONto unseen obstacles and appearance changes, we first had RECON explore in a new "junkyard" to learn to reach a goal image containing a blue dumpster (Fig. 5-a). Then, without any more exploration, we evaluated the learned goal-reaching policy when presented with *previously unseen* obstacles (trash cans, traffic cones, and a car) and and weather conditions (sunny, overcast, and twilight). Fig. 5 shows trajectories taken by the robot as it successfully navigates to the goal in scenarios with varying obstacles and lighting conditions. These results suggest that the learned representations are invariant to visual distractors that do not affect robot's

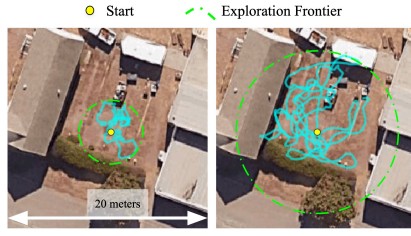

Figure 6: **Exploration via sampling** from our context-conditioned prior (*right*) allows the robot to explore 5 times faster than using random actions, e.g. in ECR [20] (*left*).

| Method | Expl. Time (mm:ss) ↓ | Nav. Time (mm:ss) ↓ | SCT [58] ↑ |
|---|---|---|---|
| Reactive | 11:54 | 00:37.4 | 0.63 |
| Random Actions | 14:54 | 00:31.4 | 0.73 |
| Vanilla Sampling | 14:06 | 00:28.7 | 0.83 |
| **Ours** | **09:56** | **00:25.8** | **0.92** |

Table 2: **Ablation experiments** confirm the importance of using an information bottleneck and a non-parametric memory.

decisions to reach a goal (e.g., changes in lighting conditions do not affect the trajectory to goal, and hence, are discarded by the bottleneck).

## 5.3 Dissecting RECON

RECON explores by sampling goals from the prior distribution over state-goal representations. To quantify the importance of this exploration strategy (**Q4**), we deploy RECON to perform undirected exploration in a novel target environment *without building a graph* of the environment. We compare the coverage of trajectories of the robot over 5 minutes of exploration when: *(a)* it executes random action sequences [20], and *(b)* it performs rollouts towards sampled goals. We see that performing rollouts to sampled goals results in 5x faster exploration in novel environments (see Fig. 6).

We also evaluate several variants of RECON that ablate its goal sampling and non-parametric memory on the end-to-end task of visual goal discovery in novel environments:

- *Reactive:* our method deployed *without* the topological graph for memory.

- *Random Actions:* a variant of our method that executes random action sequences at the frontier rather than rollouts to sampled goals. This is identical to the ECR baseline described in Sec. 5.1.

- *Vanilla Sampling:* a variant of our method which learns a goal-conditioned policy and distances *without* an information bottleneck to obtain compressed representations.

We deploy these variants in a subset of the unseen test environments and summarize their performance in Table 2. These results corroborate the observations in Fig. 6: learning a compressed goal representation is key to the performance of RECON. "Vanilla Sampling", despite sampling from a joint prior, performs poorly and is unable to discover distant goals. We hypothesize that our method is more robust because the information bottleneck helps learn a representation that ignores task-irrelevant information. We also observe that "Reactive" experiences a smaller degradation in exploration performance, suggesting that goal-sampling can help with the exploration problem even without the graph. However, we find a massive degradation in its ability to recall previously discovered goals, suggesting that the memory is key to the navigation performance of RECON.

## 6 Discussion

We proposed a system for efficiently learning goal-directed policies in new open-world environments. The key idea behind our method is to use a learned goal-conditioned distance model with a latent variable model representing visual goals for rapid goal-directed exploration. The problem setup studied in this paper, using past experience to accelerate learning in a *new* environment, is reflective of real-world robotics scenarios: collecting new experience at deployment time is costly, but experience from prior environments can provide useful guidance to solve new tasks.

In future work, we aim to provide theoretical guarantees for when and where we can expect stochastic policies and the information bottleneck to provide efficient exploration. One limitation of the current method is that it does not explicitly account for the value of information. Accounting for such states can generate a better goal-reaching policy.

## Acknowledgments

This research was supported by ARL DCIST CRA W911NF-17-2-0181, DARPA Assured Autonomy, and the Office of Naval Research. The authors would like to thank Suraj Nair and Brian Ichter for useful discussions, and Gregory Kahn for setting up the infrastructure used for autonomous collection of real-world data.

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
