# OpenReview forum: "Rapid Exploration for Open-World Navigation with Latent Goal Models"
_robot-learning.org/CoRL/2021/Conference — CoRL2021 Oral_

### Official Review · Reviewer_7dLW · 2021-07-22

**Originality:** Very Good
**Technical Quality:** Good
**Clarity Of Presentation:** Very Good
**Impact:** 4

**Recommendation:**

Strong Accept: I recommend accepting the paper and will argue for my recommendation even if other reviewers hold a different opinion.

**Summary:**

This paper presents a robot learning system for goal-directed exploration in novel environments where the robot is tasked to navigate to a goal region in the environment while provided with a goal image taken at the goal region. The system, called Rapid Exploration Controllers for Outcome-driven Navigation (RECON) consists of two parts: An autoencoder that serves as the policy and distance estimator, given the goal image and the current image of the robot, and a topological map that is constructed online to guide the robot towards exploring the environment efficiently. The system is evaluated and compared to 6 baseline methods on a set of previously unseen real-world scenarios.

**Issues:**

The points above regarding unclear explanations and missing notations need to be addressed.

**Reviewer Expertise:**

Good: General knowledge of the area

**Strengths And Weaknesses:**

Overall, the method is interesting, albeit somewhat incremental, and the motivations behind the two parts of the system (i.e. the autoencoder and the topological map) are clear. The paper does a good job in specifying the problem it attempts to address. A big plus of the system is the fact that it doesn't require any geometric map of the environment, which is often unavailable or hard to obtain for complex outdoor scenarios. The experimental results indicate that the system is more efficient in exploring previously unseen environments and reaching a goal within the environment compared to the baseline methods. I also appreciate the fact that the system was tested on a physical robot, which indicates that the system is effective in real-world scenarios.

The main problem I have with the paper is that some concepts and ideas are not very well explained which makes the paper hard to follow at times, particularly in Section 4.2:

First, the graph (i.e. the "topological memory") and what it represents needs to be explained, idealy as early as Section 3 instead of handwavy introducing it in Section 4.2. Additionally, I was initially under the impression that the nodes in the graph represent states and the edges represent their connectivity, but after examining the algorithms, it seems that the nodes represent images and the edges represent distances. This needs to be mentioned.

Second, in Section 4.2 (i) (Feasible Goal), it is not clear to me how a large probability of the goal embedding z_g^t under the prior implies that the model believes it can reach the goal?

Third, in Section Section 4.2 (ii) the paper mentions that the robot determines if it is at the frontier of the graph by checking it's distance to the least explored node within a distance threshold. How does this work? Line 8 in Algorithm 1 seems to suggest that the mean of the decoder is used to measure the distance between o_t (the current observation) and the observation associated to a node (I suppose this is what o_n denotes in Algorithm 1?). However, I'm just guessing here because this is not explained in the paper.

Fourth, how and when exactly is the graph constructed? Appendix B1 mentions that at the end of each trajectory, the subroutine ExpandGraph is called to do this. So I guess this routine should be called in the for loop in Algorithm 3?

Additionally, the paper should mention the structure of the trajectory dataset that was used to train the model in Section 4.1. Equation 2 reveals that a datapoint consists of a observation-goal observation-action-distance quadruple, but this is nowhere specified in the main text.

UPDATE:
Based on the author's feedback and the revisions that were carried out, I'm changing my recommendation from "Weak Accept" to "Strong Accept".

**Summary Of Recommendation:**

The paper proposes a system for for visual-guided goal discovery that seems to work well in practice. However, some revisions are neccessary to improve the clarity of the paper.

---

> ### Author Response · Authors · 2021-08-22
> **Response to Reviewer 7dLW: implementation details and improving clarity of presentation**
>
> The reviewer's main concerns seemed to be about the presentation of ideas and lack of implementation details in Section 4.2. As described below, we have significantly revised the paper to improve presentation and address these changes (changes in the PDF are highlighted in red). Together with answers to specific reviewer concerns below, we believe that this addresses the reviewer's concerns. We ask that the reviewer revisit their review in light of these changes, and encourage them to raise additional concerns they may have.
>
>
> ***1. [Interpretation of the topological map]***
> The second interpretation is correct: nodes represent image observations and edges represent distances. We have updated the manuscript (changes in red) at several places, including as early as the abstract, Figure 2 and Section 4 , to clarify this.
>
>
> ***2. [Clarifying feasibility of goals (Section 4.2 (i))]***
> A large probability under the prior implies that the model is confident of its distance estimates and policy, and hence, can reach the nearby goal with a high probability of success. We have modified the text to clarify this better.
> During training, the model is updated such that context-goal pairs ($o_t, o_g$) that were observed during the training get assigned a higher probability under the prior $r(.)$ (which is a unit Gaussian). Intuitively, since the network is only trained to predict distances and actions on context-goal pairs that were within the same trajectory and close to each other (or “positives”), an unseen context-goal pair is _out of distribution_ for the network and hence, have a low probability under the prior. This is useful because the estimates from the distance function and policy of the network can be unreliable for OOD inputs and this probability under the prior gives us an uncertainty estimate.
>
> ***3. [Clarifying the frontier of exploration (Section 4.2 (ii))]***
> There are two steps here: (i) we use the $LeastExploredNeighbor$ [Alg. 3] subroutine, which is a heuristic that uses the visitation counts of the neighbors to find the least explored node in the local neighborhood; let this neighbor be $o_n$ (this is an image observation, since nodes are images), and (ii) we query the distance function using the current observation ($o_t$) and $o_n$. Since the decoder is variational, we use the mean of the distance as an estimate of the distance and ignore the variance. $\delta_1$ is a small distance threshold of 4m [Table 3]. We have updated Section 4.2 (ii) to enunciate this better.
>
> ***4. [Graph construction and update]***
> In the target environment, the graph is generated by incremental updates made according to [Alg. 1]. We initiate RECON in the target with an empty graph [Alg. 1, Line 2]; after completing a trajectory to subgoal, the graph is updated with the endpoint of the trajectory [Alg. 1, Line 14] (highlighted in red text). The $ExpandGraph$ subroutine is called *after* the execution of $SubgoalNavigate$.
>
> ***5. [Structure of the trajectory dataset]***
> Thank you for pointing this out, we have mentioned this in Section 4.1.
>
> Please let us know if these modifications address your concerns regarding our submission. We will make more global edits to improve the clarity of presentation and readability after addressing comments from all reviewers.

---

> > ### Comment · Reviewer_7dLW · 2021-09-02
> > **Response to response**
> >
> > Dear Authors,
> >
> > Thank you for your detailed response. From what I can see, all of my concerns have been adequately addressed and the (already strong) paper has been further improved. Based on this, I'm willing to raise my score from "Weak Accept" to "Strong Accept". I think the paper will make for a nice contribution to the conference.
> >
> > I have only one more (extremely minor) point: Could you use slightly lower-resolution images for Figure 2? Page 2, which includes Figure 2, renders a bit slow on my machine. There are good open source programs you can use to compress images without affecting the visual quality, such as Gimp.
> >
> > - Reviewer 7dLW

---

> > > ### Author Response · Authors · 2021-09-02
> > > **Thank you!**
> > >
> > > Thank you very much, we appreciate your feedback! We'll update the figures as suggested.

---

> ### Author Response · Authors · 2021-08-26
> **Checking back**
>
> We just wanted to check back if the reviewers had any updated comments or concerns that can help us improve the quality and presentation of our submission. While we have a few more days of discussion period left (due Aug 30, Monday), we would love to discuss and make changes that help us improve the work and the reviewer's recommendation.

---

### Official Review · Reviewer_fB76 · 2021-07-23

**Originality:** Very Good
**Technical Quality:** Excellent
**Clarity Of Presentation:** Excellent
**Impact:** 4

**Recommendation:**

Strong Accept: I recommend accepting the paper and will argue for my recommendation even if other reviewers hold a different opinion.

**Summary:**

The paper proposes an approach to learn a latent representation of relative goals to be able to produce actions and distance estimates from a current image and a goal image. This model, trained on offline data, is used to build a topological map of a new environment, explore rapidly this new environment, and later navigate in this environment. Performances evaluations and comparisons with a large set of baselines are provided on a real robot in several outdoors environments, showing improved exploration and navigation speed.


**Issues:**

See the minor weaknesses mentioned above.

**Reviewer Expertise:**

Very good: Comprehensive knowledge of the area

**Strengths And Weaknesses:**

The paper is very well written, the motivation and relation to the state of the art are clear, the technical description is sufficient in the paper and well complemented in the appendix and supplementary video. At a high level, the proposed approach illustrates the well-known fact that goal-based exploration is much more efficient than random action sampling, but its implementation as a vision-based latent goal model seems an interesting contribution and its evaluation on real robots in challenging outdoor environments is quite impressive.

I don't see any major problem with the paper, I have only a few questions that could be clarified :
- What is the reward function used in the PPO baseline? Only sparse at the goal or using a more informative reward?
- Could you illustrate the interest of fine-tuning the model? Is it important in the performances? And does the model specialize in the environment or integrates knowledge and remains valid for other environments (i.e. no forgetting)?
- Line 195: you mention a relabeling scheme that is not described in sec 3.2 nor in the appendix.

**Summary Of Recommendation:**

The paper proposes an interesting new approach, with a solid experimental evaluation.

---

> ### Author Response · Authors · 2021-08-22
> **Response to Reviewer fB76: implementation details**
>
> Thank you for your encouraging comments and feedback. We provide answers to your questions below:
>
> > What is the reward function used in the PPO baseline?
>
> We use a sparse reward (+1 at the goal, 0 otherwise) with a discount factor of 0.99, which is standard in RL literature.
>
> ---
> > What is the importance of fine-tuning the model?
>
> Fine-tuning our model in the target environment is important for allowing our method to learn environment-specific information and generalize to new environments.
> Training across various environments helps our learned models to be applicable and accurate on a wide variety of (unseen) environments, which is essential to the exploration and goal-reaching performance of the model.
>
> ---
> > details about the relabeling scheme
>
> We use the self-supervised data labeling scheme proposed in BADGR [1] to automatically generate and label the trajectories using events from onboard sensors. The event labels (collision, bumpiness and position) are described in Appendix A.1 [Line 500] and more information about this pipeline can be found in [BADGR](https://sites.google.com/view/badgr). We have updated the manuscript to clarify this (highlighted in green text).
> For all of our reported experiments, we only use the collision labels to truncate trajectories into collision-free paths (and for generating rewards in RL baselines like PPO).
>
> ---
>
> _References:_
> [1] G. Khan, P. Abbeel, and S. Levine, “BADGR: An Autonomous Self-Supervised Learning-Based Navigation System” (2020)

---

> > ### Comment · Reviewer_fB76 · 2021-08-31
> > **Updated review**
> >
> > The answers to my questions  and the general discussion on the other comments are convincing for me, I confirm my first review.

---

### Official Review · Reviewer_nTAe · 2021-07-24

**Originality:** Good
**Technical Quality:** Very Good
**Clarity Of Presentation:** Very Good
**Impact:** 3

**Recommendation:**

Weak Accept: I recommend accepting the paper, but will not argue for my recommendation if the majority of other reviewers have a different opinion.

**Summary:**

The paper addresses the problem of finding a goal image specified by a user in a possibly unknown environment using a single ground mobile robot. The paper proposes a method that uses learned goal-conditioned distance model with a latent variable model representing visual goals, trained on a variety of environments, to explore the environment, and uses a topological graph to navigate towards the found goal. The paper shows experimental results with a real ground robot in known and unknown environments to validate the proposed method.

**Issues:**

The revised paper could discuss the questions raised in the "Strengths and Weaknesses" box.

**Reviewer Expertise:**

Very good: Comprehensive knowledge of the area

**Strengths And Weaknesses:**

The paper presented an interesting learning system which improves the state of the art in image-goal navigation by handling new environments and at the same time using prior experience in other environments. The paper is generally well written, with provided motivations behind the choices made. The paper is technically sound with a rather complete experimental evaluation with a real robot that includes comparisons with other 6 methods in different settings and an ablation study. It is quite impressive the dataset that was collected over a span of 18 months.

The main comments about the paper revolve around the actual practicality of the proposed method and the lack of comparison with alternative non-learning-based approaches. Some of the questions include: in outdoor, what are some of the actual uses of the proposed method which does not use any other sensor other than camera, as GPS is typically reliable, especially considering the types of environments considered in the paper? While different environments have been considered, the environments share similar characteristics, i.e., some human-made structures with large passages; how does the method generalize to totally different environments, e.g., in forest? What happens when the goal is much further away than 80m? What is the motivation to use a pure learning-based approach for the full pipeline, when for example visual SLAM could be used to create a map; the robot could follow a coverage pattern; and eventually find the goal avoiding to have paths that look suboptimal? Discussions on these aspects would improve the quality and the impact of the paper.

A minor comment regards the dataset used, which could be better described in terms of how the 20 hours of offline data.


**Summary Of Recommendation:**

While the paper could discuss the practicality and compare with more traditional exploration systems, the paper presents an interesting method for image-goal navigation in unseen environments with a rather complete evaluation that includes a real robot.

---

> ### Author Response · Authors · 2021-08-22
> **Response to Reviewer nTAe: discussion on practicality and adding new experiment [1/2]**
>
> The reviewer's main concerns seemed to be (1) on the choice of a pure learning-based pipeline and relying solely on image observations, and (2) the performance of our method on truly out-of-distribution environments. Below, we discuss how images are an intuitive way to specify semantic goals and our motivation for a learning-based pipeline. We also share results from deploying our method in a forest environment (as the reviewer requested) and provide a new video of the experiment. We acknowledge the importance of this discussion and will include it in the final submission after addressing all comments. Together with answers to specific reviewer concerns below, we believe that this addresses the reviewer's concerns. We ask that the reviewer revisit their review in light of these changes, and encourage them to raise additional concerns they may have.
>
> ---
> > The environments share similar characteristics; how does the method generalize to totally different environments, e.g. in a forest?
>
> The paper already contains experiments in two environments that are drastically different from training: farmland and urban sidewalks (both of which are far away from where the robot was trained). While we did not emphasize this difference in this submission, [we include photos here](https://drive.google.com/drive/folders/1HrMpEOHxVOf1VpelEVGLg70K0ckZG5XF?usp=sharing ) that illustrate how different these settings are. The environments are presented in Table 1, Section 5.
> Furthermore, we ran the forest experiment as suggested by the reviewer. [Here’s a video of the robot navigating across a dense patch of trees to find the goal (a cabin)](https://drive.google.com/file/d/1om1qnoaE3kT6N0GA9boncjdlhf9nXaQy/view?usp=sharing). Note that the robot has never been trained in this environment, or any environment with a dense growth of trees. Qualitatively, the model shows effective generalization and is successful in adapting itself to novel environments using the data collected in the exploration phase. During the ongoing rebuttal period, we did not have time or infrastructure to run all the comparisons in this environment, but we confirmed (as shown in the video) that our system is effective at reaching the goal. We can add more rigorous quantitative evaluations as well, but hopefully this initial qualitative experiment provides a reasonable illustration.
>
> ---
> > “GPS is typically reliable in the types of environments considered in the paper”
>
> While GPS is certainly a useful signal for localization in outdoor environments and has enabled a variety of outdoor navigation applications over long distances, it is not very convenient to specify semantic goals with GPS. The capability of operating with visual goals allows us to specify high-level directives such as “go to the blue house” or “find the dumpster”, just like you would describe semantic waypoints to a human being, which can be useful for tasks like inspection and off-road navigation in unmapped areas like deserts. Furthermore, preliminary experiments (not reported in the paper) on extending RECON by fusing visual observations with GPS localization and we find that using GPS to guide the exploration can greatly accelerate the process.
> We do not claim that navigating solely using egocentric images is the ideal way to explore novel environments, but we posit that it is an intuitive mechanism to specify goals for human users and provide a practical, sample-efficient way to utilize large amounts of autonomously collected visual navigation data by leveraging the representational capabilities of learned latent models. Utilizing structured information from other sensors, such as GPS, can certainly be powerful but a rigorous study of alternative sensor suites is not in the scope of our work.
>
> ---
> > What happens when the goal is further away?
>
> When the goal is further than $100$m away, RECON fails to find the goal in $40$ minutes and running experiments on the robot for any longer duration is infeasible. However, this is not surprising since the robot does not have a map of the environment and (directed) exploration in a novel environment is a difficult task. While this is feasible in structured environments (for example, when the robot is in a hallway and can explore along it), it is quite challenging in open environments and we see the alternative methods struggling to discover goals over $20$m away in the larger environments [D-H, Figure 8 in Appendix A.1].
>
> ---

---

> > ### Author Response · Authors · 2021-08-22
> > **Response to Reviewer nTAe: discussion on practicality and adding new experiment [2/2]**
> >
> > > What is the motivation to use a pure learning-based approach for the full pipeline?
> >
> > While there are a number of reasons why learning-based methods may be preferred over classical approaches like SLAM and feature-based tracking, a full comparison to the wide range of different methods proposed in the literature is outside the scope of this work. Such a comparison would be very difficult, since such methods generally have different assumptions and often use a very different (and well-equipped) sensor suite. If we were to add a comparison to a specific monocular SLAM approach, for instance, it would be easy to criticize the work for low-balling the comparison or not representing the (very wide) field of monocular SLAM fairly. Further, there is evidence [1] that visual SLAM in off-road environments struggles with reasoning about traversability (for example, tall grass is traversable but it often appears as an obstacle) and we faced these challenges when trying to implement a monocular SLAM baseline, particularly in the forest-type environments.
> >
> > Therefore, we believe that the scientifically sound thing to do is to scope our claims around learning-based robotic navigation, which itself is a broad field, and provide a thorough set of comparisons to other learning approaches that have addressed similar problems.
> >
> > ---
> > > More details on how the 20 hours of offline data was collected.
> >
> > The data used for pre-training our models was collected over the course of 18 months and re-uses a large amount of data collected by our collaborators for their independent projects. A large amount of data was collected by using the self-supervised data collection strategy proposed by Kahn et al. [1], followed by automated relabeling using collision labels: this consists of data collected using a time-correlated random walk in the real-world with minimal human supervision. The dataset also includes roughly 1.5 hours of human-teleoperated data of the robot in a university campus and roughly 3 hours of data collected on city sidewalks and parks using a learned policy (developed by our collaborators). We have now released a subset of the dataset, along with visualization scripts [at this link](https://drive.google.com/drive/folders/1L6vyycGkLXHS_yCbFoj0hbbPL30l_AYt?usp=sharing)
> >
> > We do not provide references, locations and further details on this in the appendix for these works in order to preserve the anonymity of the review process but will share these details in the final version of the paper. We also discuss more information about the dataset in response to reviewer gi9D and you can find more details there.
> >
> > _References:_
> > [1] G. Khan, P. Abbeel, and S. Levine, “BADGR: An Autonomous Self-Supervised Learning-Based Navigation System” (2020)

---

> > ### Comment · Reviewer_nTAe · 2021-08-31
> > **Great the inclusion of a new experiment and discussion about practicality**
> >
> > Thanks to the authors for the thorough response provided. The responses are reasonable: while in practice other sensors could be used to accelerate the exploration as noted in the response, the paper contribution remains solid, considering the scope of the paper and the additional discussion together with the experiments in a totally different type of environment that are promised to be included in the revised paper. Thus, the review remains positive.

---

> ### Author Response · Authors · 2021-08-26
> **Checking back**
>
> We just wanted to check back if the reviewers had any updated comments or concerns that can help us improve the quality and presentation of our submission. While we have a few more days of discussion period left (due Aug 30, Monday), we would love to discuss and make changes that help us improve the work and the reviewer's recommendation.

---

### Official Review · Reviewer_gi9D · 2021-07-25

**Originality:** Good
**Technical Quality:** Good
**Clarity Of Presentation:** Excellent
**Impact:** 4

**Recommendation:**

Weak Accept: I recommend accepting the paper, but will not argue for my recommendation if the majority of other reviewers have a different opinion.

**Summary:**

Full disclosure; I was made aware of this paper by [name removed] tweeting about it in April: [link removed] and I’m also aware that this work was presented at an ICLR workshop this year: [link removed]. So at best, this is a single-blind review.

The paper presents a novel method for goal-driven visual outdoor navigation. The method is pretrained on a large dataset to propose goals for navigation and the online algorithm creates a graph of the environment and learns compressed visual embeddings that are robust to weather, light conditions, and distractor objects.

The work is overall very impressive and I respect the sheer amount of field experiments that were carried out for this work. However, I think that the work currently suffers from some clarity issues to the point of being impossible to reproduce without contacting the authors.



**Issues:**

See section "Weaknesses" above. I numbered all my issues with this paper so the authors can respond in turn. Some additional questions, mostly out of interest:

8. What's the formula for SCT in table 1?
9. You mentioned moving objects (people) but I didn't see any in your Figures and video. How does your model work when there's foot traffic, qualitatively?

**Reviewer Expertise:**

Good: General knowledge of the area

**Strengths And Weaknesses:**

### Strengths

- The content that made it into the paper is well-written and well-presented. The text itself is clear, the Figures are illustrative, the pseudocode blocks are mostly great, and the experimental section is excellent.
- As mentioned above, the experimental time commitment in this work is staggering. The authors mentioned over 100 hours of real-world experiments and there was probably also robot maintenance and intermittent tuning of the method, so I have to commend the authors for that sheer effort!
- I really appreciated that the authors clearly stated their research questions at the beginning of the experiments section.
- The choice of recent baselines and amount thereof is great.

### Weaknesses

- Practicality: The authors mentioned that this was only possible by having a large offline dataset of the robot and also all the data is collected on the same robot platform. That raises many questions that weren't addressed in the paper:
  1. You emphasized that the training and testing environments weren't the same. But how disjoint/out-of-distribution is the data? What's the closest distance between any possible sample from the train and test dataset. I feel like if they were recorded in the same neighborhoods, the test environment is very much in-distribution and some of the claims of generality need to be toned down a bit.
  2. I'm unclear what the importance of the perspective is in this method - can the data be recorded from a different perspective (height above ground) or with a different visual sensor? What about a different FOV?
  3. Part of your motivation is that this method uses "only" real-world data and doesn't need to be pretrained in simulation (see related works section line 66 and following) but also your "...method enables the robot to explore an environment from scratch in just 20 minutes, using prior experience from other environments". What about pretraining your method in simulation? Have you looked into that? Would any of this work if the dataset was simulated? Because I feel like a large dataset for pretraining would be easier to come by if it were simulated.
  4. You also mention in the main body of the paper that you're making the dataset publicly available (in the appendix?) (line 109) but I see no evidence of that. The dataset could have been uploaded to some anonymous google drive folder or cloud storage to be given to the reviewers for anonymous review.
- Reproducibility: The present text raises many questions about how to reimplement this:
	5. You mention in the main body of the paper that the data is collected over 18 months through random walk but in the appendix, you further specify that the data is not random but obtained from a heuristic that resets after collision and also something about bumpiness and self-labeling. The reader is left utterly lacking any understanding of this dataset. (a) How many data points are there? (b) What is the distribution of the data, i.e. are there any over- or underexplored areas, what is the distribution of "bumpinesses" and collisions in your data? (c) did you preprocess the data in any way? (d) Did you filter the data to exclude certain episodes or events?
	6. How exactly is the pretraining done? (a) I.e. what parts are pretrained and for how long? (b) How is the prior $r(\cdot)$ calculated?
	7. In Algorithm 1, there are a few unclear things - some of which are explained in the appendix. (a) the function LeastExploredNeighbor is taking a hyperparameter $\delta_2$ and the purpose of this HP is unclear. This is explained in the appendix but from my best understanding, the appendix is there to provide additional examples, and additional clarification but the main body of the paper should be able to be clear and self-contained. (b) Same for the function SubgoalNavigate and the hyperparameter $H$. (c) What is $o_n$ in line 11? (d) If $r(\cdot)$ is the prior over goals that returns the value of a normal distribution (i.e. a scalar, line 128), then how is a goal sampled from that in line 9?



**Summary Of Recommendation:**

I think this paper has a lot of potential and I would like to see this accepted but to recommend acceptance, I first need some crucial details clarified (see weaknesses section above).

---

> ### Author Response · Authors · 2021-08-22
> **Response to Reviewer gi9D: releasing dataset, implementation details, discussion on practicality [1/3]**
>
> The reviewer's main concerns seemed to be about the lack of implementation details for reproducibility and on the practicality of the setup considered. As described below, we have significantly revised the paper to address these changes (changes in the PDF are highlighted in magenta). We have released a subset of the dataset (as requested by the reviewer) and also shared new video results from a forest environment that is very different from the training environments, showing good generalization by RECON. Together with answers to specific reviewer concerns below, we believe that this addresses the reviewer's concerns. We ask that the reviewer revisit their review in light of these changes, and let us know if there are any other concerns.
>
> While we strongly agree with the reviewer that providing ample details to make the work maximally reproducible is important, and we hope that the additional details we provided below help in that regard, we want to also emphasize that CoRL has an 8-page limit, and no matter how much we might want to include all relevant details in the main paper, we are constrained by the page limit. Therefore, much of this material needs to go in an appendix. But we are happy to include everything possible in the appendix, and would appreciate any further suggestions about what else would help to improve reproducibility.
>
> ***4. [Making the dataset public.]***
> For immediate perusal, we are releasing [a subset of the dataset](https://drive.google.com/drive/folders/1L6vyycGkLXHS_yCbFoj0hbbPL30l_AYt?usp=sharing), containing 800 trajectories collected over 2.5 hours of interaction in real-world environments. We are also providing visualization scripts to browse the trajectory dataset.
> We are prepared to release the data but need to figure out an effective way to host it (the dataset is over 80GB), which is difficult to do anonymously.
>
>
> ***5. [Details about the collection and composition of the dataset.]***
> The data is indeed collected using a time-correlated random walk, giving us “smooth” random trajectories in the environment. To automatically label the data, we use collision detectors (either using an inexpensive 2D LiDAR or the IMU to detect “stuck” events) to generate collision labels; these labels are used to dissect the random walks into smooth trajectories that end in collision (hindsight relabeling). Since there are no arbitrary resets in the real-world, we also program a backup maneuver that can move the robot away from an obstacle once the collision is detected: this allows the robot to operate with minimal human supervision (except in irrecoverable cases like flipping). This self-supervised data collection and labeling scheme was first proposed by Kahn et al. [1]. We have modified the text in the appendix to clarify this and added more information about the dataset (Section A.1).
> - a) [How many data points are there?]
> The dataset contains 5000 trajectories, which can be used to generate upto 4M data points for training the model (note that each data point is a start-goal pair).
> - b) [What is the distribution of the data?]
> The dataset contains trajectories from 9 different environments (as shown in the Figure 7). Each trajectory is at least 20 seconds long and 40% of the trajectories end in collision; a small number of trajectories (~2%) end in irrecoverable states such as the robot flipping or wheels getting stuck in mud.
> - c) [Did you preprocess the data?]
> The only preprocessing done on the trajectories is the labeling scheme described earlier, which uses collision labels to truncate the trajectories into smaller collision-free segments. The dataset contains raw measurements from a wide range of onboard sensors including a pair of stereo RGB cameras, thermal camera, 2D LiDAR, GPS and IMU to support offline evaluation using an alternative suite of sensors.
> - d) [Did you filter the data?]
> No.
>
>
> ***6. [How is the pre-training done?]***
> The entire model is trained with the offline trajectory dataset, which was collected over a multitude of environments as discussed above. This allows the model to reason about general navigational affordances, semantics and traversability; the model is finetuned in the target environment using data collected during exploration/deployment, which allows it to adapt to the structure and contents of the specific environment. The model is trained to convergence, which occurs in about 15-20 epochs of training, but depending on the size and diversity of the dataset (we ran experiments in early stages of the project with subsets and different environments) this number can vary anywhere between 5 and 40 epochs. Following the commonly used VIB and VAE frameworks [2, 3], we fix $r(\cdot)$ to be the standard Gaussian distribution $\mathcal{N}(0, I)$.

---

> > ### Author Response · Authors · 2021-08-22
> > **Response to Reviewer gi9D: releasing dataset, implementation details, discussion on practicality [2/3]**
> >
> > ***7. [Clarifications in the algorithm.]***
> > - a) [LeastExploredNeighbour]
> > Thank you for pointing this out, we have updated Section 4.2 (ii) to improve the presentation of the algorithm (text highlighted in magenta & red, since another reviewer also pointed this out).
> > - b) [SubgoalNavigate]
> > We have updated Section 4.2 and Appendix B.1 to improve clarity.
> > - c) [Alg. 1, Line 11]
> > $o_n$ is the least explored neighbor, as returned in line 4.
> > - d) [Sampling from the prior $r(\cdot)$]
> > The latent $z^g_t$ is a vector (for instance, 64-dim in our case). Hence, the distribution r(.) is a multivariate (standard) normal distribution, chosen to have mean 0 and variance identity $I$. This enables us to compute the KL between $p_\phi$ and $r$ [Eq. 2] in the VIB loss [3] and sample from the distributions in Alg. 1 [Lines 5, 9, 11]. In Line 9, this amounts to sampling a random $z^w_t$ from a multivariate (standard) normal distribution.
> >
> >
> > ***1. [How disjoint are the training and test data?]***
> > The training and test locations differ by being collected in different locations between 250m and 12km apart. For the core results of the paper (Section 5.1, Table 1), the training data was collected in the warehouse (Figure 7(c)), parking lot 1 (Figure 7(e)), and residential (Figure 7(i)) environments, along with some data collected on city sidewalks (figure not in paper to preserve anonymity). The methods were evaluated on environments of varying size; the test environments include (non-exhaustive) the parking lot 2 (Figure 7(h), 500m away from the training parking lot but similar structure), junkyard (Figure 7(a), 300m away to the nearest training location, but with novel objects like dumpsters and trash cans that were absent from the training set), cafeteria (Figure 7(d), 1km from nearest training location and with novel objects like patio tables) and forest cabin (Figure 7(f), 12km away from all training locations and a very different appearance). The test environments were chosen to be of varying complexity to effectively compare the performance of the different methods (see Figure 8).
> >
> > Additionally, we ran our system in more drastically different settings. [Here’s a video of our method deployed in the forest cabin environment (the goal is the cabin)](https://drive.google.com/file/d/1om1qnoaE3kT6N0GA9boncjdlhf9nXaQy/view?usp=sharing).
> > Note that the robot was trained on data from a suburban area with buildings and has never seen an environment like this. It’s difficult to characterize differences in environments more rigorously (as there is no set measurement for this), so we believe this qualitative characterization is the best we can do, but we would appreciate any concrete feedback or suggestions you might have for how to improve this.
> >
> > ***2. [What is the importance of perspective or camera parameters?]***
> > This is a great question! As an artifact of our dataset being collected over multiple projects and collaborators, our dataset consists of images from 2 different camera sensors and captured from at least 3 different heights (not entirely by choice; there was no calibration done and the camera locations were 3-5cm apart). While a rigorous study of the robustness to camera parameters is outside the scope of our work, we expect the models (encoders consisting of convolutional layers) to be fairly robust to small affine perturbations.
> >
> > ***3. [What about pre-training in simulation?]***
> > We would like to emphasize that a key contribution of this work is to demonstrate learning of meaningful goal-driven behaviors from a large amount of unrelated, offline dataset of experience and minimal human supervision. Our method is largely agnostic to whether the data is real or simulated, and any sim-to-real method could be used in combination with our approach too, but using real data has many advantages, particularly in open-world environments that are difficult to model in simulation and present a large sim-to-real gap. Our preliminary experiments in a photorealistic outdoor navigation simulator failed to transfer the learned policies to the real-world -- likely due to the difficulty in simulating phenomena like bumpy terrain, moving through bushes, etc. It’s quite possible that simulated training can work, but this is not in the scope of our work -- the paper is already quite dense, and this would distract the reader from our main contributions.
> >
> > ***8. [What is the formula of SCT in Table 1?]***
> > SCT = $s \frac{t’}{max(t, t’)}$, where $s$ is 0 or 1 indicating success in reaching the goal, $t’$ is the shortest possible amount of time it takes to reach the goal from the start position. We obtain this by having a human “expert” teleoperate the robot.  robot. $t$ is the time taken by the robot to reach the goal. A detailed motivation and analysis of this metric can be found in Yokoyama et al. [4]

---

> > > ### Author Response · Authors · 2021-08-22
> > > **Response to Reviewer gi9D: releasing dataset, implementation details, discussion on practicality [3/3]**
> > >
> > > ***9. [Dynamic obstacles]***
> > > We study robustness to settings where obstacles are located in different positions. Obstacles do not move within a single episode, but may change positions between the exploration and navigation phases. Our experiments do not consider “dynamic obstacles” within a trajectory and we have revised Section 5.2 to clarify this.
> > >
> > >
> > > _References:_
> > > [1] G. Khan, P. Abbeel, and S. Levine, “BADGR: An Autonomous Self-Supervised Learning-Based Navigation System” (2020)
> > > [2] D. Kingma, and M. Welling, “Auto-Encoding Variational Bayes” (2013)
> > > [3] A. Alemi, I. Fischer, J. Dillon, and K. Murphy, “Deep Variational Information Bottleneck” (2016)
> > > [4] N. Yokoyama, S. Ha, and D. Batra, “Success Weighted by Completion Time: A Dynamics-Aware Evaluation Criteria for Embodied Navigation” (2021)

---

> > > > ### Comment · Reviewer_gi9D · 2021-08-30
> > > > **good work, increasing score**
> > > >
> > > > The additional implementation details, esp. Algo.3 and Tab.4 in the appendix, and the dataset sample are highly appreciated. based on these, I have increased my score.

---

> > > > > ### Author Response · Authors · 2021-08-31
> > > > > **Thank you!**
> > > > >
> > > > > We thank the reviewer for re-evaluating the paper. If the reviewer has any additional concerns that would help improve the quality of the submission and/or improve their recommendation, we would be happy to discuss and address them here.

---

> ### Author Response · Authors · 2021-08-26
> **Checking back**
>
> We just wanted to check back if the reviewers had any updated comments or concerns that can help us improve the quality and presentation of our submission. While we have a few more days of discussion period left (due Aug 30, Monday), we would love to discuss and make changes that help us improve the work and the reviewer's recommendation.

---

### Meta-Review · Area_Chair_JCDD · 2021-08-13

**Recommendation:** Accept (Oral)
**Confidence:** 5

**Metareview:**

In this paper, a robotic learning system for goal-oriented exploration in a novel environment is proposed. A compact representation of goal images is learned and a topological memory is constructed online incrementally during the exploration. All reviewers recognize the field experiments conducted. The concerns are mostly focused on the practicality of the proposed system and some more detailed explanation are lacking.

In the rebuttal session, the authors have responded to the reviewers' concerns and more details are provided. And the reviewers have reached a consensus of accepting this paper.

---

> ### Author Response · Authors · 2021-08-22
> **Response to Area Chair JCDD**
>
> Thank you for the feedback! The main concerns raised by the reviewers can be grouped as: (1) experiments in more drastically different environments, (2) lack of details about the dataset and public release, (3) missing details in the presentation that make it difficult to reproduce the method (practicality & reproducibility). We’ve addressed all of these, as summarized below.
>
> 1. As suggested by reviewer nTAe, we compiled additional experiments that demonstrate generalization to more drastically different settings like forests. We share [a new video of the robot navigating across a dense patch of trees to find the goal (a cabin)](https://drive.google.com/file/d/1om1qnoaE3kT6N0GA9boncjdlhf9nXaQy/view?usp=sharing)
> We have also updated the paper to discuss in more detail how the existing training and test environments differ (see also the response to Reviewer nTAe).
>
> 2. For immediate perusal, we are [releasing a subset of our trajectory dataset](https://drive.google.com/drive/folders/1L6vyycGkLXHS_yCbFoj0hbbPL30l_AYt?usp=sharing), containing 800 trajectories collected over 2.5 hours of interaction in real-world environments. We are also providing a visualization script to browse the trajectory dataset.
> We are prepared to release the data but need to figure out an effective way to host it (the dataset is over 80GB), which is difficult to do anonymously, as we would need to use our own servers. Additionally, we have also shared more details about dataset composition and distribution in response to the concerns raised by reviewer gi9D.
>
> 3. We have modified the paper significantly (changes highlighted in red/magenta/green text) to address the concerns about clarity of presentation. While we strongly agree that providing ample details to make the work maximally reproducible is important, and we hope that the additional details we provided below help in that regard, we want to also emphasize that CoRL has an 8-page limit and no matter how much we might want to include all relevant details in the main paper, we are constrained by the page limit. Therefore, much of this material had to go in an appendix and we have tried our best to include more information to improve reproducibility. We are happy to include everything possible in the appendix, and would appreciate any further suggestions about what else would help to improve reproducibility.
>
> Reviewer nTAe asked about why we use a learning-based system (as opposed to a non-learning-based system): we would emphasize here that we do not claim that our learning-based approach is the best possible way to address such tasks -- comparing learning vs. non-learning methods is outside the scope of our work. While there are a number of reasons why learning-based methods may be preferred over classical approaches like SLAM and feature-based tracking, a full comparison to the wide range of different methods proposed in the literature is very difficult to do due to differences in assumptions and the wide range of non-learning methods. Therefore, we believe that the scientifically sound thing to do is to scope our claims around learning-based robotic navigation, which itself is a broad field, and provide a thorough set of comparisons to other learning approaches that have addressed similar problems.

---

### Decision · Program_Chairs · 2021-09-13

**Decision:**

Accept (Oral)

**Comment:**

In this paper, a robotic learning system for goal-oriented exploration in a novel environment is proposed. A compact representation of goal images is learned and a topological memory is constructed online incrementally during the exploration. All reviewers recognize the field experiments conducted. The concerns are mostly focused on the practicality of the proposed system and some more detailed explanation are lacking.

In the rebuttal session, the authors have responded to the reviewers' concerns and more details are provided. And the reviewers have reached a consensus of accepting this paper.